# Multimodal Fusion via Hypergraph Autoencoder and Contrastive Learning for Emotion Recognition in Conversation

## ABSTRACT

Multimodal emotion recognition in conversation (MERC) seeks to identify the speakers' emotions expressed in each utterance, offering significant potential across diverse fields. The challenge of MERC lies in balancing speaker modeling and context modeling, encompassing both long-distance and short-distance contexts, as well as addressing the complexity of multimodal information fusion. Recent research adopts graph-based methods to model intricate conversational relationships effectively. Nevertheless, the majority of these methods utilize a fixed fully connected structure to link all utterances, relying on convolution to interpret complex context. This approach can inherently heighten the redundancy in contextual messages and excessive graph network smoothing, particularly in the context of long-distance conversations. To address this issue, we propose a framework that dynamically adjusts hypergraph connections by variational hypergraph autoencoder (VHGAE), and employs contrastive learning to mitigate uncertainty factors during the reconstruction process. Experimental results demonstrate the effectiveness of our proposal against the state-of-the-art methods on IEMOCAP and MELD datasets. We release the code to support the reproducibility of this work (currently it is uploaded as the "complementary material" within the review system and will be made public following the completion of the review process).

## CCS CONCEPTS

• **Information systems** → **Sentiment analysis**; • **Computing methodologies** → **Discourse, dialogue and pragmatics**.

## KEYWORDS

Multimodal Emotion Recognition in Conversation, Variational Hypergraph Autoencoder, Contrastive Learning, Multimodal Fusion

## 1 INTRODUCTION

Emotion is one of the crucial characteristics of human behavior [19]. Experienced psychiatrists can assess emotions by observing an individual's behavior, which serves as a key indicator for understanding their inclinations and responses. As human-computer interaction (HCI) advances, the capability to discern emotions from dialogues using multimodal information is becoming increasingly significant [29, 38]. This process is commonly referred to as multimodal emotion recognition in conversation (MERC). The multimodality

herein includes different modal information such as the speaker's language, tone, facial expression, body movement and so on [6, 35]. From a modeling perspective, a conversation consists of a sequence of utterances. Each utterance contains one or more modalities of information and is linked to speaker information. The target of MERC is to identify the emotion category of each utterance by analyzing the available information and contextual cues.

Compared with the emotion recognition in non-dialogue scenarios [10], MERC necessitates a specific emphasis on modelling the speakers involved in the dialogue. Also unlike the analysis of single-modal information [9], the processing of multimodal information demands the utilization of distinct processing techniques to extract meaningful information from various modalities. Different modalities of information need to be synthesized to facilitate the comprehensive analysis of a conversation. For example, when a speaker utters the word "ok" with a tone of helplessness, solely relying on textual cues may not fully convey the speaker's emotional state. By taking into account factors such as intonation and tone assist in inferring the underlying feeling of sadness expressed by the speaker. Efficient integration and utilization of multimodal information play a crucial role in enhancing the precision of emotion recognition during conversations [31].

Current research methodologies regarding MERC can be classified into two main categories: non-graph-based method [13, 24, 25] and graph-based method [3, 9, 12, 14, 21]. **Non-graph-based method** typically utilizes recurrent neural networks (RNN) or long short-term memory (LSTM) to capture contextual information, while the output utterance representations are used for label classification. However, these methods encounter challenges in modeling long-range dependencies because of issues in information propagation and gradient vanishing problems [9]. **Graph-based method** typically uses a graph to depict a conversation, with each utterance represented as a node and the relationships between utterances shown through edge weights or connections between nodes. Contextual information is captured through graph convolutions, and the resulting node embeddings are fed into subsequent classification steps [19].

Graph-based methods can be further divided into standard graph-based and hypergraph-based methods. **Standard graph-based method** [9, 12, 14, 21] is a typical graph-based method, which represents textual information in utterances as nodes and captures contextual relationships by connecting nodes with various types of edges within a specific window size. For the standard graph-based methods, the pairwise connection approach fails to depict the actual physical structure of MERC accurately. Additionally, as the number of graph convolution layers rises, the training time and storage requirements increase exponentially. It can also result in oversmoothing of the graph and redundancy of nodes, potentially leading to inaccurate assessments [27].

**Hypergraph-based method** changes the point-to-point connection to a hyperedge connection structure that more closely fits the model [3]. Hypergraph is a special graph structure capable of capturing high-order correlations, enabling the exploration of more intricate relationships [1]. By linking multiple modalities within a single utterance and connecting all nodes of the same modality using hyperedges, the hypergraph-based method can achieve outstanding performance improvement. Nevertheless, the fixed fully connected hypergraph structure still results in information redundancy, graph smoothing and slow convergence, especially when processing long-distance conversations [37].

To address the aforementioned issues in existing hypergraph-based methods, we propose a multimodal fusion framework via hypergraph autoencoder and contrastive learning named **HAUCL** for MERC, which is applicable to multimodal data and capable of adaptively adjusting hypergraph connections. The framework consists of five modules: (1) unimodal encoding, (2) hypergraph construction, (3) hypergraph convolution, (4) hypergraph contrastive learning, and (5) classifier. The **unimodal encoding** module is designed to generate modality-independent representations. For the **hypergraph construction** module, it firstly forms an initial fully connected hypergraph structure. Then, a variational hypergraph autoencoder (VHGAE)-based approach is introduced to realize adaptive adjustment of the hypergraph. In this paper, we develop VHGAE to map the hypergraph to the latent space to obtain node and hyperedge by sampling from space, and then learn new connections via Gumbel-Softmax [16]. The aforementioned procedures exhibit a degree of randomness. To minimize the influence of random factors, two parameter-sharing paths are established through the utilization of contrastive learning techniques: Two VHGAEs reconstruct the hypergraph, and the reconstructed hypergraphs are utilized in the subsequent **hypergraph convolution** module to learn the embeddings along with contextual information. Then, point-to-point **hypergraph contrastive learning** module is applied to the obtained two hypergraphs, where nodes corresponding to each other in different hypergraphs are considered positive sample pairs to ensure model stability. Conversely, other nodes are treated as negative sample pairs to enhance the learning of more distinctive embeddings. Finally, the learned embeddings are fed into the **classifier** module for emotion category prediction.

The main contributions of this paper are summarized as follows:

- We propose a joint learning framework based on hypergraphs, which achieved synergistic optimization of hypergraph reconstruction, contrastive learning, and emotion recognition, leading to globally optimal performance. Specifically, VHGAE is integrated into MERC to adaptively adjust the hypergraph, while Gumbel-Softmax is devised to mitigate data overflow.
- We utilize contrastive learning to mitigate the impact of uncertainty in the sampling process and the Gumbel-softmax learning process of VHGAE, enhancing the robustness and stability of the model.
- Extensive experiments conducted on two mainstream MERC datasets, IEMOCAP and MELD, validate the effectiveness of our work. The results showed that our proposal performed

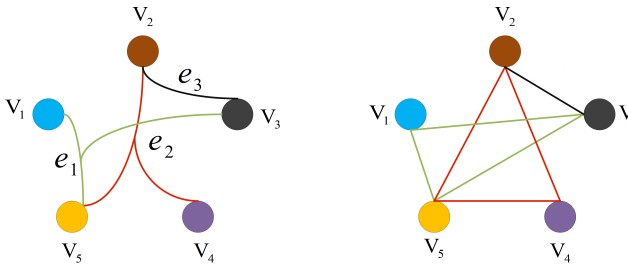

**Figure 1: An illustration showcasing the differences between hypergraphs (left) and standard graphs (right).**

superiorly compared to the state-of-the-art methods in accuracy and weighted F1 score.

## 2 RELATED WORK

### 2.1 Multimodal Emotion Recognition

Regarding the non-graph learning methods, BC-LSTM captures contextual information from surrounding utterances in three different modalities by using three independent bidirectional LSTM networks, and the output utterance representations are used for label classification [25]. However, this method lacks the usage of speaker information and thus is not applicable to multi-person conversation scenarios. DialogueRNN utilizes three gate recurrent units (GRUs) to track the global context, speaker state, and emotion state throughout the entire dialogue, which effectively integrates speaker modeling, contextual modeling, and emotion modeling [24]. To mimic the human reasoning process, DialogueCRN introduces reasoning modules to integrate the factors that make emotions happen [13].

The standard graph-based methods, such as DialogueGCN [9], represent textual information in utterances as nodes and capture contextual relationships through different types of edges connecting nodes within a given window size. The multimodal graph convolutional network (MMGCN) develops DialogueGCN by further incorporating audio and video modalities into the model [14]. To address the challenge of cross-modal interaction in information fusion within ERC, MIMMN introduces a multi-view network that leverages complementary information from all modalities. It dynamically balances the relationships between all modalities during the fusion process [32]. MM-DFN [12] uses a dynamic fusion mechanism to fully understand the context relationship between multiple modalities and reduce the redundancy between modalities. COG-MEN [17] utilizes graph neural network (GNN) to leverage both local and global information in a conversation. GraphMFT [21] not only designs a multimodal fusion method based on graphs but also utilizes multiple graph attention networks (GATs) to capture the intra-modal contextual details and inter-modal complementary information. M3NET [3] introduces the hypergraph into the field of MERC. Through simple fully connected structures and randomly initialized edge weights, significant improvement in prediction accuracy and time efficiency has been achieved by multiple hypergraph convolutions.

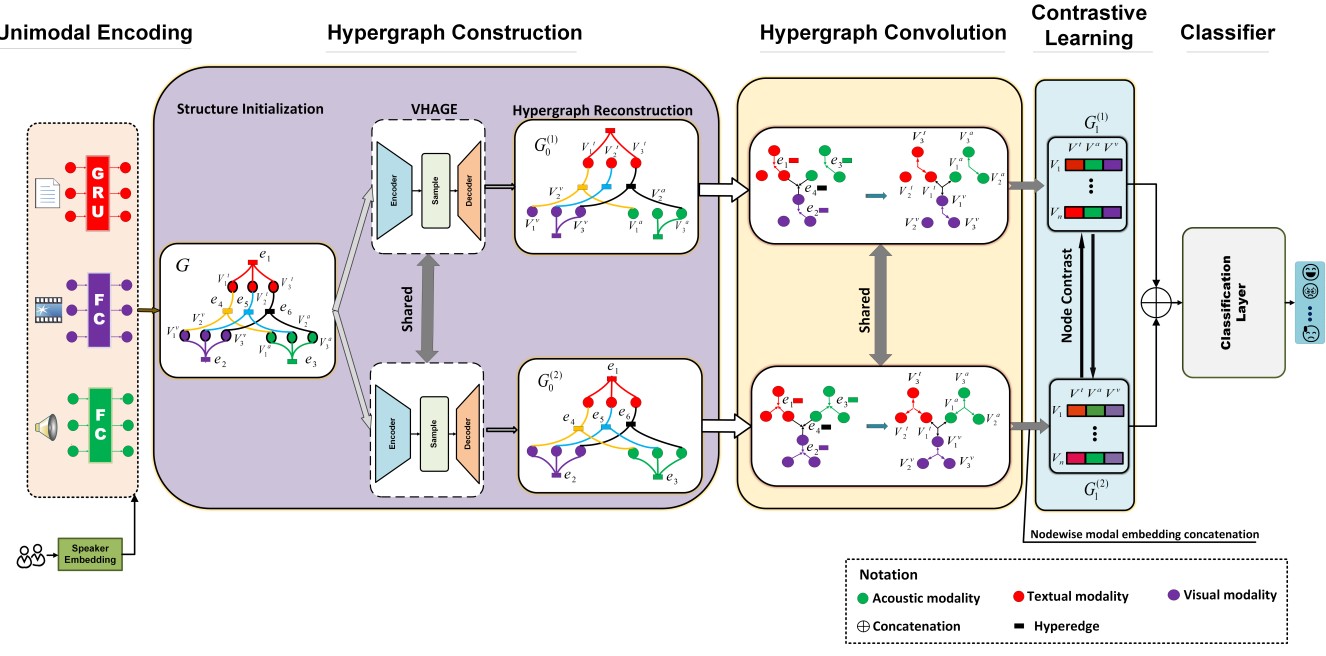

Figure 2: An overview of our proposed framework HAUCL.

## 2.2 Hypergraph Learning

A hypergraph acts as an extended version of the standard graph learning, specifically designed to extract high-order correlations within the data [1]. The examples of hypergraphs are shown on the left side of Figure 1, with the corresponding standard graphs shown on the right side. The circular dots represent five nodes, i.e., from $V_1$ to $V_5$. Curves with the same color form a hyperedge, and there are three hyperedges $e_1, e_2, e_3$ in total. In a hypergraph, connections are not limited to pairwise relationships as in a standard graph. Hyperedges can link multiple nodes together, and a single node can be linked by multiple hyperedges simultaneously. Meanwhile a hypergraph can include multiple types of hyperedges, representing multiple meanings. In this paper, we create a hypergraph in which all utterances linked to the same speaker are grouped together on a hyperedge, while also connecting similar modality into another hyperedge. This structure closely resembles the physical structure of certain models, capturing higher-order correlations and minimizing information loss during the modeling process. The effectiveness of hypergraph learning in solving the association problem of multimodal data has already been verified in various applications, such as including recommendation system [33], video segmentation [34], sleep stage classification [23], and drug-target interaction prediction [28].

Regarding hypergraph convolutions, the learning process involves aggregating node information onto connected hyperedges with varying weights, followed by sending messages from the hyperedges back to the connected nodes. This process is not constrained by distance, thereby mitigating the limitations of message transmission during the process [7]. These benefits are particularly pronounced in long-distance transmissions [8]. Therefore, the hypergraph learning process is anticipated to be effective in the MERC task, as speakers frequently discuss topics that are distant from the current conversation, utilizing long-distance cues.

## 3 METHODOLOGY

In this paper, we model the MERC task as follows: a conversation contains a sequence of utterances $u_i$ ($i = 1, ..., N$). $N$ is the number of utterances. Each utterance $u_i$ consists of textual, acoustic, and visual modality, represented as $u_i = \{u_i^t, u_i^a, u_i^v\}$, respectively. Meanwhile, each $u_i$ is spoken by a corresponding person $s_i$. By integrating the speaker information, an utterance can be denoted as $v_i = (u_i, s_i)$. The goal of a MERC task is to predict the emotion label for each utterance $v_i$ based on the given multimodal information. The overall framework of our proposed HAUCL is illustrated in Figure 2. It includes unimodal encoding, hypergraph construction, convolution, contrastive learning and classifier.

### 3.1 Preprocess and Unimodal Encoding

This module involves extracting essential information from raw visual, textual, and acoustic modalities data. Following the approach outlined in M3NET [3], features from visual modalities are extracted using DenseNet [15] or 3D-CNN [36], depending on the adopted dataset. Features from acoustic and textual modalities are extracted using the OpenSmile toolkit [5] and the RoBERTa large model [22] respectively.

As mentioned above, incorporating contextual information is crucial for emotion category prediction in conversations. To enhance discourse feature representation, we employ various encoding methods tailored to the characteristics of different modalities.

Specifically, we utilize GRU network [4] to encode context information for the textual modality, while acoustic and visual information is encoded using two fully connected multilayer perceptrons (MLPs). To facilitate the information fusion across modalities, we normalize the encoded dimension to a unified $d$ dimension as below:

$$U_i^a = W_a u_i^a + b_i^a \tag{1a}$$

$$U_i^v = W_v u_i^v + b_i^v \tag{1b}$$

$$U_i^t = W_t \left( \overleftrightarrow{GRU}(U_{i-1}^t, u_i^t, u_{i+1}^t) \right) + b_i^t \tag{1c}$$

where $u_i$ is the input of unimodal encoding. $U_i$ is the output of the model with the dimension $d$. $a, v, t$ stands for visual, acoustic, and textual modalities respectively. $W$ and $b$ are trainable parameters.

Speaker information is a critical factor that affects the performance of the MERC task. We firstly encode the speaker information into vectors $s_i$ in one-hot form as:

$$S_i = W_s s_i + b_i^s \tag{2}$$

Next, we integrate them into the modality information by:

$$V_i^x = S_i + U_i^x, x \in \{t, a, v\} \tag{3}$$

The output of this module is $V_i$, which is the feature embeddings with modality-independent context awareness and speaker information.

## 3.2 Hypergraph Construction

This module is composed of three stages: structure initialization, VHGAE, and hypergraph reconstruction.

*3.2.1 Structure Initialization.* We represent a conversation with continuous utterances through hypergraph $\mathcal{G} = (\mathcal{V}, \mathcal{E})$, where each node $v \in V$ represents a unimodal utterance and each hyperedge $h \in H$ captures multimodal dependencies. Each utterance's modality is represented by a node in a hypergraph, i.e., $V_i^t$ for the textual modality, $V_i^a$ for the acoustic modality, and $V_i^v$ for the visual modality.

We design two distinct types of hyperedges in this paper: the first one involves connecting every node in a modality to form a hyperedge , i.e., it includes $\{V_1^v, V_2^v, ..., V_N^v\}$, $\{V_1^t, V_2^t, ..., V_N^t\}$, and $\{V_1^a, V_2^a, .., V_N^a\}$. The second type of hypergraph creates a hyperedge $\{V_i^a, V_v^t, V_i^t\}$ by joining the three modalities of an utterance.

Similar to the standard graphs, the incidence matrix for hypergraphs can also be defined as $\mathcal{H} \in \mathbb{R}^{3N \times M}$, where $N$ and $M$ is the number of nodes and hyperedges, respectively. We define $H_{i,j}$ to determine the presence of node $i$ in hyperedge $j$ as:

$$H_{i,j} = \begin{cases} 1, & \text{node } i \text{ is included in hyperedge } j \\ 0, & \text{otherwise} \end{cases} \tag{4}$$

*3.2.2 VHGAE.* The fully connected hypergraph generated in the structure initialization stage may lead to redundancy in the subsequent update process, impeding the classification of subtle differences. To mitigate this challenge, we introduce VHGAE to reconstruct the hypergraph, aiming to identify the most appropriate hypergraph structure. VHGAE comprises of three processes: encoder, sampler, and decoder. The structure of VHGAE is illustrated in Figure 3.

**Encoder**: It aims to project the hypergraph into a representation consisting of sets of nodes and hyperedges. This projection can

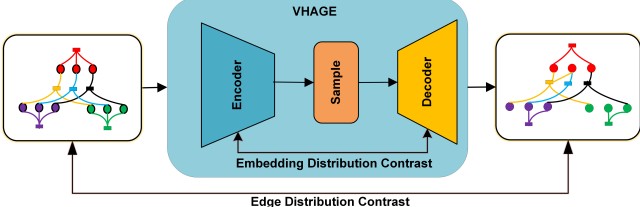

**Figure 3: The structure of VHGAE.**

facilitate the subsequent decoding process of the non-Euclidean structure, as highlighted in the original paper that proposed variational graph auto-encoders (VGAE) [20].

In our proposed method, we follow the VHGAE framework and utilize a hypergraph neural network (HyperGNN) to perform hypergraph convolution on the original hypergraph. This convolution operation produces embeddings for both the nodes $v$ and the hyperedges $\epsilon$ as:

$$v, \epsilon = \text{HyperGNN}(\mathcal{G}) \tag{5}$$

We utilize the obtained embeddings to encode the mean $\mu$ and variance $\sigma$ vectors for each type $k \in (v, \epsilon)$. This encoding process involves applying linear transformations and activation functions, as described by the following equations:

$$\mu_k = W_1^k \left( \sigma(W_\mu^k(k) + b_\mu^k) \right) + b_1^k \tag{6a}$$

$$\sigma_k = \sigma_1 \left( W_2^k (\sigma(W_\sigma^k(k) + b_\sigma^k)) + b_2^k \right) \tag{6b}$$

where $W^k$ and $b^k$ are learnable parameters specific to the type $k$. The activation functions $\sigma$ and $\sigma_1$ correspond to the ReLU and Softplus functions, respectively.

Through encoding the mean and variance vectors with node and hyperedge embeddings, we can effectively capture and represent the crucial information regarding the hypergraph's structure within a latent space. These encoded vectors will play a pivotal role in the subsequent stages, enabling the generation of meaningful and relevant outputs.

**Sample**: To incorporate the reparametrization trick, we utilize sampling in the latent space to obtain new nodes and hyperedge embeddings. The sampling process introduces stochasticity while ensuring differentiable computations during the training phase.

To generate the new embeddings, we use the mean $\mu_k$ and variance $\sigma_k$ vectors obtained from the encoder process by Equations 6a and 6b. The reparametrization trick involves sampling from a standard normal distribution $\delta \sim N(0, 1)$ and scaling it by the standard deviation $\sigma_k$. The obtained sample is then added element-wise to the mean vector $\mu_k$ to obtain the new embedding $m_k$ by:

$$m_k = \mu_k + \sigma_k \odot \delta \tag{7}$$

where $\odot$ represents the element-wise product between $\sigma_k$ and $\delta$. By incorporating the sampled noise $\delta$ into the latent space representation, we introduce randomness to the model, while maintaining differentiability for efficient optimization.

The obtained embeddings $m_k$ serve as the updated representations for the output nodes or hyperedges, capturing the variability and uncertainty within the hypergraph structure.

 

**Decoder**: This process aims to reconstruct the hypergraph from the latent space representation. By leveraging the updated embeddings obtained from the encoder process, we can recover the connection structure of the new hypergraph through a series of operations.

First, we calculate the matrix $h_i$ by taking the dot product between the transpose of $m_\epsilon$ and $m_\sigma$ by:

$$h_i = m_\epsilon^T m_\sigma \qquad (8)$$

where $m_\sigma^T$ represents the inverse of $m_\sigma$.

Next, we apply the Gumbel-Softmax function to the matrix $h$ with a temperature coefficient $\tau$ to introduce stochasticity:

$$h = \text{softmax}\Big(\text{Gumbel\_Softmax}(h_i, \tau) + p\Big) \qquad (9)$$

In the above equation, Gumbel_Softmax is a function that applies the Gumbel-Softmax relaxation. To prevent data overflow, we incorporate the addition of a constant $p$ to the Gumbel-Softmax operation and subsequently apply the softmax function.

After conducting the softmax operation, The obtained matrix $h$ has two columns, representing a distribution over the hypergraph connections. We extract the first column of matrix $h$, which corresponds to the incidence matrix of the new hypergraph $\mathcal{G}_0 = (\mathcal{V}, \mathcal{E}_0)$.

By obtaining the connection structure of the new hypergraph through the decoder, we can reconstruct the relationships and connections between nodes and hyperedges. This reconstructed hypergraph can then be further utilized for various downstream tasks.

**Loss function**: The loss function in VHGAE consists of primary components designed to produce a reconstructed hypergraph that closely resembles the original one. Specifically, the first component measures the Kullback-Leibler (KL) divergence between the distributions of the latent variables (nodes and hyperedges) and their corresponding prior distributions. The second component quantifies the difference in connection structure between the newly generated hypergraph and the original hypergraph. The VHGAE's loss function $\mathcal{L}_g$ is defined as follows:

$$\mathcal{L}_g = \text{KL}(m_\sigma, \sigma) + \text{KL}(m_\epsilon, \epsilon) + \text{CE}(h_0, h) \qquad (10)$$

where $\text{KL}(m_\sigma, \sigma)$ measures the KL divergence between the distribution of the sampled latent variables $m_\sigma$ and the prior distribution $\sigma$. Similarly, $\text{KL}(m_\epsilon, \epsilon)$ represents the KL divergence between the distribution of the sampled hyperedge embeddings $m_\epsilon$ and the prior distribution $\epsilon$. The third term $\text{CE}(h_0, h)$ denotes the cross-entropy loss function, quantifying the connection structure difference between the original hypergraph $h_0$ and the generated hypergraph $h$. This component ensures that the generated hypergraph closely matches the original hypergraph regarding the distribution of connections.

By minimizing this loss function, VHGAE aims to learn an effective latent space representation that captures the essential characteristics of the hypergraph while preserving its connection structure.

### 3.3 Hypergraph Convolution

With the new hypergraph $\mathcal{G}_0 = (\mathcal{V}, \mathcal{E}_0)$, we first perform node convolution by aggregating node features to update the hyperedge embeddings. The aggregation stage facilitates the integration of information from neighboring nodes into the hyperedge representation. Following the update of the hyperedge embeddings, we proceed to the hyperedge convolution stage, where hyperedge messages are disseminated to the nodes. This operation enables the information propagation from hyperedges to their incident nodes. For each hyperedge $\epsilon \in \mathcal{E}_0$, we aggregate the embeddings of its incident nodes $v$ according to a predefined aggregation function by:

$$n_\epsilon = \text{Agg}\Big(\{n_v\}_{v \in \epsilon}\Big) \qquad (11)$$

where $n_\epsilon$ represents the updated embedding for the hyperedge $\epsilon$ and $n_v$ denotes the embedding of the node $v$. The aggregation function Agg combines the embeddings of the incident nodes to generate the new hyperedge embedding. For each node $v \in \mathcal{V}$, we aggregate the messages from its incident hyperedges $\epsilon$ using a predefined aggregation function similar to Equation 11.

By performing the node and hyperedge convolutions, we can effectively propagate information and update the embeddings in the hypergraph $\mathcal{G}_1 = (\mathcal{V}_0, \mathcal{E}_0)$. This reformulated solution enables the capture of the relationships and interactions between nodes and hyperedges, facilitating a more comprehensive understanding of the hypergraph structure.

### 3.4 Hypergraph Contrastive Learning

In order to mitigate the instability inherent in the sampling and decoding processes, we devise a dual-path scheme within our model. The primary objective is to minimize the dissimilarity between corresponding points in two hypergraphs $\mathcal{G}_1^{(1)} = (\mathcal{V}_0^{(1)}, \mathcal{E}_0)$ and $\mathcal{G}_1^{(2)} = (\mathcal{V}_0^{(2)}, \mathcal{E}_0)$, which are obtained through the progression of VHGAE and convolution. Concurrently, we aim to maximize the distance between each point and other points within the embedding space.

Within the context of the two hypergraph views, pairs of vertices that correspond to one another are regarded as positive pairs, whereas the remaining vertex pairs are considered negative pairs. The embedding of the $i$-th vertex in the two views is denoted as $v_i^{(1)} \in \mathcal{V}_0^{(1)}$ and $v_i^{(2)} \in \mathcal{V}_0^{(2)}$. The contrastive loss $\mathcal{L}_{cl}$ between $\mathcal{V}_0^{(1)}$ and $\mathcal{V}_0^{(2)}$ is:

$$\mathcal{L}_{cl} = \frac{1}{2|\mathcal{V}_0|} \sum_{i=1}^{|\mathcal{V}_0|} \Big(f(v_i^{(1)}, v_i^{(2)}) + f(v_i^{(2)}, v_i^{(1)})\Big) \qquad (12)$$

Here, $\mathcal{V}_0$ denotes the set of vertices and $|\mathcal{V}_0|$ signifies the cardinality of $\mathcal{V}_0$. The term $f(v_i^{(1)}, v_i^{(2)})$ is calculated as:

$$f(v_i^{(1)}, v_i^{(2)}) = -\log\left(\frac{q(v_i^{(1)}, v_i^{(2)})}{q(v_i^{(1)}, v_i^{(2)}) + \sum_{i \neq j} q(v_i^{(1)}, v_j^{(2)}) + \sum_{i \neq j} q(v_i^{(1)}, v_j^{(1)})}\right) \qquad (13)$$

where

$$q(x, y) = e^{\frac{g(x,y)}{\tau}} \qquad (14)$$

Here, $\tau$ is a temperature parameter and $g(,)$ denotes the cosine similarity function. Considering that the function $g(,)$ is not symmetric, we average the positive and negative aspects. Specifically, $\sum_{i \neq j} q(v_i^{(1)}, v_j^{(2)})$ and $\sum_{i \neq j} q(v_i^{(1)}, v_j^{(1)})$ denote the negative pairs

in the other graph and the same graph, respectively. Meanwhile, $q(v_i^{(1)}, v_i^{(2)})$ represents a positive pair in the other graph.

By minimizing the combined loss function $\mathcal{L}_{cl}$, the similarity between corresponding points is expected to increase while enhancing the distance between each point and other points within the embedding space. This approach promotes alignment and discrimination of the embeddings, thereby yielding more stable and meaningful representations of the hypergraph structure.

## 3.5 Emotion Classifier

After acquiring contextual knowledge, we perform a fusion process on the node embeddings of the two hypergraphs $\mathcal{G}_1^{(1)}$ and $\mathcal{G}_1^{(2)}$ to obtain $\mathcal{G}_2 = (\mathcal{V}_2, \mathcal{E}_2)$. This process aims to integrate the information from the two hypergraphs into a unified representation.

Following that, we concatenate the node embeddings of the three modalities that belong to the same utterance, resulting in a comprehensive representation. Specifically, let $\{v_i^t, v_i^a, v_i^v\} \in \mathcal{V}_2$ denote the node embeddings of the hypergraphs corresponding to the textual, acoustic, and visual modalities, respectively. We concatenate these embeddings to obtain a fused representation by:

$$v_i = \text{Concatenate}(v_i^t, v_i^a, v_i^v) \tag{15}$$

The Concatenate function combines the embeddings of the three modalities into a unified vector, allowing for the integration of multiple sources of information. The fused representation $v_i$ encompasses a broader range of information, enhancing subsequent analysis and prediction tasks by providing a more comprehensive input.

Given the fused representation $v_i$ for an utterance, the formulas for predicting the emotion label are as follows:

$$\widehat{v_i} = ReLU(W_2 v_i + b_2) \tag{16a}$$

$$P_i = \text{softmax}(W_3 \widehat{v_i} + b_3) \tag{16b}$$

$$\widehat{y_i} = \text{argmax}(P_i[\tau]) \tag{16c}$$

In these formulas, $W_2$ and $W_3$ are weight matrices, $b_2$ and $b_3$ are bias vectors, $\widehat{v_i}$ is the processed output of $v_i$ using the ReLU activation function, $P_i$ is the probability distribution over the emotion labels, and $\widehat{y_i}$ is the predicted emotion label. $\tau$ represents the dimension corresponding to the emotion labels. By applying these formulas, we can predict the emotion label for each utterance based on the fused representation $v_i$ and the learned parameters $W_2$, $W_3$, $b_2$, and $b_3$.

## 3.6 Training Objectives

We use categorical cross-entropy loss with $L2$ regularization term to define the error loss between the predicted emotion category and the true label during the training process as below:

$$\mathcal{L}_{ce} = -\frac{1}{\sum_{s=1}^{N} c(s)} \sum_{i=1}^{N} \sum_{j=1}^{c(i)} \log P_{i,j}[y_{i,j}] + \lambda \|\theta\|_2 \tag{17}$$

where $N$ represents the number of dialogues in a dataset. $c(s)$ represents the number of utterances in dialogue $s$. It is worth noting that each dialogue can have a different number of utterances. $P_{i,j}$ denotes the predicted probability distribution of emotion labels for utterance $j$ in dialogue $i$, while $y_{i,j}$ represents the expected class

**Table 1: Main hyperparameters for HAUCL.**

| Dataset | Batch size | Learning rate | $\lambda_g$ | $\lambda_{cl}$ | Epoch | Dropout |
|---------|-----------|---------------|-------------|----------------|-------|---------|
| MELD    | 12        | 0.0001        | 0.5         | 1              | 15    | 0.4     |
| IEMOCAP | 12        | 0.0001        | 0.8         | 0.1            | 45    | 0.3     |

label. The regularization weight $\lambda$ controls the importance of the regularization term relative to the cross-entropy loss.

By combining Equations 17, 10 and 12, we define the final loss function as:

$$\mathcal{L} = \mathcal{L}_{ce} + \lambda_g \mathcal{L}_g + \lambda_{cl} \mathcal{L}_{cl} \tag{18}$$

where the hyperparameter weights $\lambda_g$ and $\lambda_{cl}$ control the importance of the generalized adversarial loss and the contrastive loss, respectively.

## 4 EXPERIMENT

### 4.1 Datasets

In this paper, we conduct experiments on two popular multimodal datasets in the field of MERC: the interactive emotional dyadic motion capture database (**IEMOCAP**) [2] and multimodal emotionlines dataset (**MELD**) [26].

- IEMOCAP: It contains videos of two-way conversations with 10 actors (5 male and 5 female). IEMOCAP records the tone and power of speech, facial expressions, torso posture, head position, gestures, transcripts, and gaze in a duo session. In this paper, we use facial expressions, the tone and power of speech and transcripts. The emotions in this dataset are artificially classified into six categories: happy, sad, neutral, angry, excited, and frustrated. We use 120 dialogues containing 5,810 utterances for training and validation, while the remaining 31 dialogues with 1623 utterances for testing.
- MELD: It is a multimodal dataset for emotion recognition in multi-party conversations, containing textual, acoustic and visual modalities for ERC, selected from Friends TV series. This dataset includes seven emotions: neutral, surprise, fear, sadness, happiness, disgust, and anger. We use 1,153 dialogues with 11,098 utterances for training and validation, while the rest 280 dialogues with 2610 utterances for testing.

It is worth noting that IEMOCAP dataset features a fixed set of two speakers engaging in multiple rounds of conversation, whereas MELD dataset may involve multiple speakers but with fewer utterances per conversation. Meanwhile, the emotion distribution within MELD dataset is imbalanced, with a significantly higher proportion of "neutral" emotions compared to other emotional categories, comprising nearly half of the dataset. These characteristics pose significant challenges to the model's stability.

### 4.2 Experimental Settings and Baselines

We perform all experiments on an NVIDIA GTX 1050Ti with Win11 operating system. The versions of Pytorch and cuda are 2.1.2 and 11.8, respectively. Adam optimizer is used for training. We set the batch size as 12 and the learning rate as 0.0001 on both datasets. The hyperparameter $\tau$ of Gumbel-softmax in Equation 9 is 0.1 . The number of hypergraph convolutions is 1. More details regarding the main parameters can be found in Table 1

Table 2: Performance of various methods (Bold font indicates the best performance).

| Method | IEMOCAP | | | | | | | | MELD | |
| | Emotion Categories (F1) | | | | | | Overall | | Overall | |
| | Happy | Sad | Neutral | Angry | Excited | Frustrated | Acc. | WF1 | Acc. | WF1 |
|---|---|---|---|---|---|---|---|---|---|---|
| bc-LSTM [25] | 32.62 | 70.34 | 51.14 | 63.44 | 67.91 | 61.06 | 59.58 | 59.10 | 59.62 | 56.80 |
| DialogueRNN [24] | 33.18 | 78.80 | 59.21 | 65.28 | 71.86 | 58.91 | 63.40 | 62.75 | 60.31 | 57.66 |
| DialogueCRN [13] | 51.59 | 74.54 | 62.38 | 67.25 | 73.96 | 59.97 | 65.31 | 65.34 | 59.66 | 56.76 |
| DialogueGCN [9] | 47.10 | 80.88 | 58.71 | 66.08 | 70.97 | 61.21 | 65.54 | 65.04 | 58.62 | 56.36 |
| MMGCN [14] | 45.45 | 77.53 | 61.99 | 66.67 | 72.04 | 64.12 | 65.56 | 68.71 | 59.31 | 57.82 |
| DIMMN [32] | 30.2 | 74.2 | 59.0 | 62.7 | 72.5 | 66.6 | 64.7 | 64.1 | 60.6 | 58.6 |
| MM-DFN [12] | 42.22 | 78.98 | 66.42 | 69.77 | 75.56 | 66.33 | 68.21 | 68.18 | 62.49 | 59.46 |
| COGMEN [17] | 51.91 | 81.72 | **68.61** | 66.02 | 75.31 | 58.23 | 68.26 | 67.63 | 62.53 | 61.77 |
| GraphMFT [21] | 45.99 | **83.12** | 63.08 | **70.30** | **76.92** | 63.84 | 67.90 | 68.07 | 61.30 | 58.37 |
| M3NET [3] | **57.96** | 81.56 | 68.30 | 65.59 | 74.91 | 63.19 | 69.01 | 69.12 | 67.62 | 66.15 |
| HAUCL (ours) | 53.57 | 82.04 | **68.61** | 66.44 | 75.60 | **68.23** | **70.30** | **70.27** | **68.05** | **66.72** |

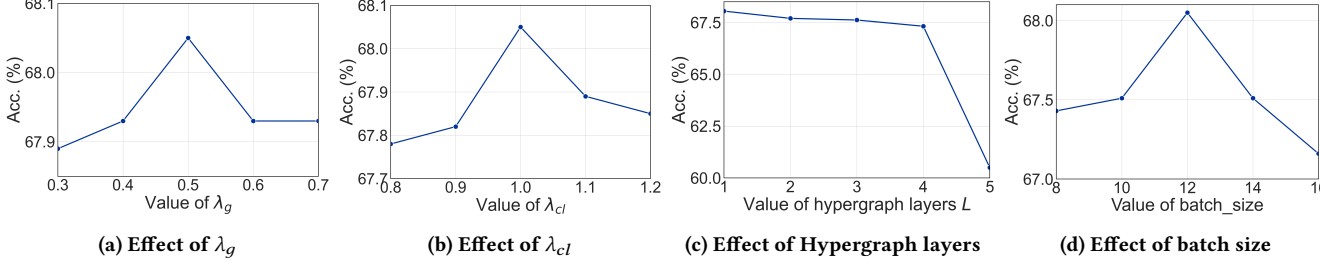

(a) Effect of $\lambda_g$  (b) Effect of $\lambda_{cl}$  (c) Effect of Hypergraph layers  (d) Effect of batch size

Figure 4: Sensitive analysis of HAUCL on MELD dataset. All experiments test the results while fixing all other parameters with the best performance.

In order to validate the performance of the proposed method HAUCL in the MERC task, we introduce the ten state-of-the-art methods for comparison: (1) non-graph learning: LSTM [25], DialogueRNN [24], and DialogueCRN [13]; (2) standard graph learning: DialogueGCN [9], MMGCN [14], DIMMN [32], MMDFN [12], COGMEN [17] and GraphMFT [21]; and (3) hypergraph learning: M3NET [3]. More details regarding the baseline methods can be found in Section 2.

For validation, we adopt the most mainstream evaluation metrics in this field: accuracy (**Acc.**) and weighted F1 score (**WF1**).

### 4.3 Performance Comparison

Table 2 summarizes the performance of different methods tested on IEMOCAP and MELD datasets. The results show that our proposed HAUCL achieves superior performance in terms of the overall accuracy and weighted F1 score. In detail, compared with M3NET, which achieves the second-best performance, HAUCL enhances the accuracy and WF1 by 0.43% and 0.57% respectively on MELD dataset and by 1.29% and 1.15% on IEMOCAP dataset. The ability of HAUCL to dynamically modify the connection structure of the hypergraph helps in reducing information redundancy, particularly in IEMOCAP dataset with a high average utterance per conversation. Additionally, the use of hypergraphs helps prevent excessive smoothing, reducing the risk of excessive smoothing occurring in

standard graph-based methods. Compared with non-graph learning methods, our proposed HAUCL can demonstrate significant enhancement in long-distance information transmission and multimodal information fusion, resulting in satisfactory accuracy and weighted F1 score performance.

### 4.4 Sensitivity Analysis

We select the following four main parameters in HAUCL for sensitivity analysis tested on MELD dataset. Figures 4a and 4b show the ratio of hypergraph reconstruction loss and contrastive learning loss to the total loss, respectively. Figure 4c represents the number of convolution layers passed by the new hypergraph obtained by reconstruction to learn the contextual information. Figure 4d shows the effect of batch size. Similar trends are also observed on IEMOCAP data.

**The weight of the hypergraph reconstruction loss** $\lambda_g$: It reflects the deviation from the original graph over the total loss (See Equation 18). Higher values of $\lambda_g$ indicate that the reconstructed hypergraph closely resembles the original graph. As shown in Figure 4a, when $\lambda_g$ is set to 0.5, our method demonstrates optimal performance in accuracy. Meanwhile, deviating from this optimal value, either towards larger or smaller values, results in a decline in the overall performance.

**The weight of contrastive learning loss** $\lambda_{cl}$: Similar with $\lambda_g$, as the value of $\lambda_{cl}$ increases, the method will increasingly focus

**Table 3: Performance of HAUCL for ablation study.**

| Method | IEMOCAP | MELD |
|---|---|---|
| w/o SE | 69.19 | 67.24 |
| w/o GCL | 69.52 | 67.62 |
| w/o CL | 69.32 | 67.47 |
| HAUCL (ours) | **70.30** | **68.05** |

on the differences between the two hypergraphs derived from the two paths. Conversely, when the dissimilarity between the two hypergraphs diminishes, our proposal's capability to withstand interference strengthens. However, when this value is excessively large, it will impact the loss of emotion recognition, i.e., when $\lambda_{cl}$ exceeds 1.1, there is a degradation in accuracy performance as plotted in Figure 4b.

**Hypergraph layer** $L$: Figure 4c demonstrates that increasing the number of covolutional layers in a hypergraph does not necessarily lead to enhanced accuracy performance. A large value of $L$ not only amplifies the model's complexity and runtime, but also risks oversmoothing, potentially complicating the differentiation of emotions with similar characteristics. When $L$ is 5, there is a sharp decrease in accuracy performance, indicating an over-smoothing phenomenon.

**Batch size**: The selection of batch size is a crucial factor that impacts the performance of recognition [11, 18]. Given the non-uniform distribution of MELD dataset, employing a batch size that is too small can render the model susceptible to the interference of small samples, leading to significant gradient fluctuations and convergence challenges. Conversely, an excessively large batch size may prompt the model to overly generalize, potentially compromising accuracy. As depicted in Figure 4d, the best performance is attained with a batch size of 12.

## 4.5 Ablation Study

For a more comprehensive analysis of the effectiveness of our proposed method HAUCL, we conduct ablation experiments from three different aspects: the impact of (1) speaker embedding, (2) VHGAE and contrastive learning, and (3) contrastive learning in terms of accuracy performance. The results are summarized in Table 3.

**Impact of Speaker Embedding**: Speaker embedding can distinguish the input features from different speakers. Existing research has shown that incorporating speaker information can enhance the accuracy of emotion recognition tasks [14]. The exclusion of speaker embedding ("w/o SE" in Table 3) results in a decrease in accuracy, with a degradation of 1.11% and 0.81% observed on IEMOCAP and MELD datasets, respectively. These findings indicate that incorporating person modeling can enhance the model's performance in the MERC domain.

**Impact of VHGAE and Contrastive Learning**: We utilize VHGAE for dynamic hyperedge selection to minimize redundancy and employ contrastive learning to mitigate random errors. In Table 3, "w/o GCL" indicates the direct fusion of two hypergraph convolutions without the inclusion of VHGAE and contrastive learning module. The results demonstrate the effectiveness of our proposed HAUCL: It can improve the accuracy performance by 0.78% and 0.43% on IEMOCAP and MELD datasets respectively.

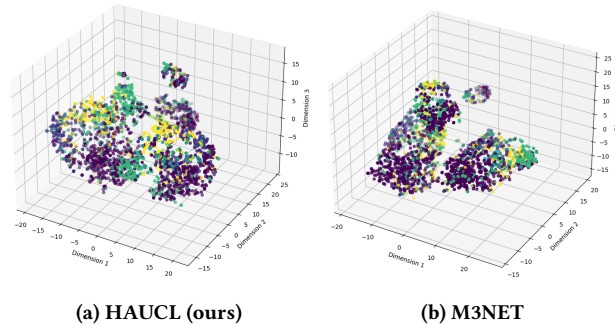

(a) HAUCL (ours)    (b) M3NET

**Figure 5: Visualization of our proposed HAUCL and M3NET on MELD dataset.**

**Impact of Contrastive Learning**: "w/o CL" in Table 3 refers to the model that incorporates hypergraph and VHGAE without the integration of contrastive learning. The experimental results indicate a 0.98% enhancement in accuracy on IEMOCAP dataset and a 0.58% improvement on MELD dataset. These results verify that the contrastive learning module effectively controls the random fluctuations of VHGAE, enhancing model accuracy while reducing information redundancy.

## 4.6 Visualization

In order to demonstrate the discriminability of nodes, we present the node representations acquired through our proposed method HAUCL and the M3Net (the second-best method in Table 2) on MELD dataset. To visualize these representations in a more comprehensive manner, we employ t-SNE [30] method for dimensionality reduction, transforming the obtained nodes into three dimensions. Furthermore, we assign distinct colors to indicate the true labels of the nodes. By comparing the two figures in Figure 5, it is evident that the data points depicted in Figure 5a (our method) exhibit greater separation, resulting in a more discriminative segmentation. As aforementioned, the representations derived from the proposed HAUCL exhibit reduced redundancy and enhanced discriminability, thereby enabling the attainment of superior outcomes.

## 5 CONCLUSION

In this paper, we propose a joint learning framework based on hypergraph learning to improve the performance of MERC. This framework aims to address the issue of excessive redundancy stemming from the fully connected structure of graphs or hypergraphs. The proposed method HAUCL effectively integrates hypergraph adaptive reconstruction and contrastive learning, which reduces information redundancy and enhances accuracy. Experimental results verify the superiority of our proposed method against state-of-the-art ones. In the future, we expect to integrate external knowledge, such as large language models (LLM), into our framework. By focusing on linear labels such as valence-arousal-dominance (VAD) in dimensional emotion space, we aim to substitute classification labels with the goal of enhancing machines' comprehension of human behavior.

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
