# OpenReview forum: "Multimodal Fusion via Hypergraph Autoencoder and Contrastive Learning for Emotion Recognition in Conversation"
_acmmm.org/ACMMM/2024/Conference — MM2024 Poster_

### Official Review · Reviewer_8Ssz · 2024-05-24

**Rating:** 3
**Confidence:** 4

**Summary:**

This paper introduces hypergraph autoencoder and contrastive learning to MERC. Experimental results on IEMOCAP and MELD demonstrate the effectiveness of this method.

**Strengths:**

Advantages:
1.	This paper is different from existing graph-based works in MERC. It combines graph-based autoencoder [1] and graph-based contrastive learning [2] and applies them to MERC.

[1] Thomas N Kipf and Max Welling. 2016. Variational graph auto-encoders. arXiv preprint arXiv:1611.07308 (2016).

[2] You, Yuning, Tianlong Chen, Yongduo Sui, Ting Chen, Zhangyang Wang, and Yang Shen. "Graph contrastive learning with augmentations." Advances in neural information processing systems 33 (2020): 5812-5823.

**Limitations:**

Disadvantages:

2.	However, this combination does not bring noticeable performance improvement. For example, in Table 2, this paper brings less 1% performance improvement for both datasets. Meanwhile, the ablation studies in Table 3 further shows that each component appears to be of little use.

3.	Please compare the training times and the model size with existing works. Maybe you can emphasize your advantages in terms of less training time and small model size rather than performance.

4.	Meanwhile, this paper draws the basic idea of graph-based autoencoder [1] and graph-based contrastive learning [2]. We would like to see some structural modifications specific to MERC rather than directly applying existing methods to MERC.

5.	There seems to be a writing error: “The second type of hypergraph creates a hyperedge {V_i^a, V_v^t, V_i^t}” => {V_i^a, V_i^v, V_i^t}

6.	Where N and M is the number of nodes and hyperedges => M=N(the number of nodes)+3 (the number of modalities)? Please specific the relationship between N and M.

[1] Thomas N Kipf and Max Welling. 2016. Variational graph auto-encoders. arXiv preprint arXiv:1611.07308 (2016).

[2] You, Yuning, Tianlong Chen, Yongduo Sui, Ting Chen, Zhangyang Wang, and Yang Shen. "Graph contrastive learning with augmentations." Advances in neural information processing systems 33 (2020): 5812-5823.

**Suitability:**

3

---

### Official Review · Reviewer_hJ4P · 2024-05-24

**Rating:** 4
**Confidence:** 3

**Summary:**

The paper proposes a novel framework to perform Multimodal emotion recognition using an hypergraph. The idea is first to encode each modality in a traditional way, then to build an hypergraph connecting nodes within each modality, and cross-modality for each utterance. A second graph is generated through a Variational Autoencoder, in order to perform some feature selection, and then the two representations are used to predict emotions for each utterance in the dialogue. Results show a better all-round performance compared to other SOTA baselines.

**Strengths:**

- The introduction clearly motivates the proposed methods and claims, providing a good overview on why the authors want to test the hypergraph idea on Multimodal Emotion Recognition.

- Autoencoding the hypergraph structure could be and interesting and novel idea, and in general the methodology section is quite clear and not too difficult to follow, despite the methods used not being straightforward.

- The benchmarking datasets used are appropriate for the task, as well as the experimental setup and evaluation metrics. I appreciate the report of the F1 score for each emotion descriptor for IEMOCAP (it should have been done for MELD too). The all-round average performance is better than other graph-based baselines.

**Limitations:**

- I find the literature review a bit lacking on recent methods based on Transformers and discriminative LLMs (such as BERT), which are indeed used here as encoders. It seems from the description that non-graph methods use mostly CNN and LSTM, but this is not true anymore recently. The authors may consider reviewing the latest SOTA methods on the analyzed datasets, for instance from paper with code benchmark lists.

- I am not sure the GRU is really needed for text encoding, as RoBERTa is already capable of capturing sequential information from the text, and can be eventually finetuned.

- In section 3.2.1, it is not really clear to me the connection rationale, why connecting every node in a modality or all the modalities of an utterances, and not for instance to connect according to other dialogue information. I also wonder whether a graph built in this way is really useful, as cross-modality information or intra-modality information can be captured by other mechanisms (for instance an attention layer). This also because, later, contrastive learning is applied between the original and reconstructed graph structures, and I wonder why similar information such as similar speakers or emotion may get distanced away as not represented in the hypergraph topology.

- Regarding the analysis, it would be better to show Figure 5 in 2D, as it is not too clear to see the class separation in 3D. I also wonder, related to the previous point and the discussion in section 4.4, what is the actual achievement of the hypergraph reconstruction, as it seems that the best performance is achieved when the reconstruction loss and the contrastive learning loss weights are set in order to enforce more close reconstruction, and therefore it is not so clear why the autoencoder at first place.

**Suitability:**

3

---

### Official Review · Reviewer_Geqp · 2024-06-02

**Rating:** 4
**Confidence:** 3

**Summary:**

The paper proposes VHGAE, Variational HyperGraph Autoencoder, to mitigate limitations of redundancy, excessive smoothing within existing graph networks while modelling long-distance dependencies across modalities to predict dominant emotion in Multimodal Emotion Recognition in conversation (MERC). The paper proposes a joint learning framework based on HyperGraphs coupled with Contrastive Learning to achieve improved performance towards MERC task.

Experiments and ablation studies were conducted on the MELD and IEMOCAP datasets.

**Strengths:**

HyperGraph Construction, HyperGraph Convolution, HyperGraph Contrastive Learning are the key contributions of the paper. Motivation and the need to capture long-distance dependencies across modalities for tasks such as MERC is well grounded. Utilizing hypergraph structure to overcome the limitation of pairwise relationships in a standard graph structure is well-motivated and well-explained in the paper.

**Limitations:**

While the HyperGraph construction aims to capture long-distance dependencies across modalities and the details of hypergraph creation are explained in the paper, there are a few questions/limitations that could be addressed.

1. Details about the computational complexity of the proposed work have not been discussed in the paper. The time taken for the proposed HyperGraph Construction (for an utterance with N feature nodes in each modality) could help understand the feasibility of the proposed work, compared to existing cross-attention based approaches.

2. Ablation studies with details about hypergraph creation in cases of corrupted/missing information within modalities can help to understand the robustness of the proposed graph-based approach.

3. Above mentioned details regarding the performance of the utilized hypergraph convolution in corrupted/missing information scenarios within modalities can help to understand the robustness of the proposed approach.

**Suitability:**

3

---

### Meta-Review · Area_Chair_DofS · 2024-07-03

**Recommendation:** Accept (Poster)
**Confidence:** 4

**Metareview:**

Test-Time Adaptation for Multimodal Sentiment Analysis